# Prediction of Incident Cancers in the Lifelines Population-Based Cohort

**DOI:** 10.3390/cancers13092133

**Published:** 2021-04-28

**Authors:** Francisco O. Cortés-Ibañez, Sunil Belur Nagaraj, Ludo Cornelissen, Gerjan J. Navis, Bert van der Vegt, Grigory Sidorenkov, Geertruida H. de Bock

**Affiliations:** 1Department of Epidemiology, University Medical Center Groningen, University of Groningen, 9713 GZ Groningen, The Netherlands; g.sidorenkov@umcg.nl (G.S.); g.h.de.bock@umcg.nl (G.H.d.B.); 2Department of Clinical Pharmacy & Pharmacology, University Medical Center Groningen, University of Groningen, 9713 GZ Groningen, The Netherlands; sbn1984@gmail.com; 3Department of Radiation Oncology, University Medical Center Groningen, University of Groningen, 9713 GZ Groningen, The Netherlands; l.j.cornelissen@rug.nl; 4Department of Internal Medicine, Division of Nephrology, University Medical Center Groningen, University of Groningen, 9713 GZ Groningen, The Netherlands; g.j.navis@umcg.nl; 5Department of Pathology & Medical Biology, University Medical Center Groningen, University of Groningen, 9713 GZ Groningen, The Netherlands; b.van.der.vegt@umcg.nl

**Keywords:** classification, prediction, neoplasms, supervised machine learning, health behavior, lifestyle

## Abstract

**Simple Summary:**

The accurate prediction of incident cancers could be relevant to understanding and reducing cancer incidence. The aim of this study was to develop machine learning (ML) models that could predict an incident diagnosis of cancer. Data were available for 116,188 cancer-free participants and 4232 incident cancer cases. The main outcome was an incident cancer (excluding skin cancer) during follow-up assessment in a population-based cohort. The performance of three ML algorithms was evaluated using supervised binary classification to identify incident cancers among participants. An overall area under the receiver operator curve (AUC) < 0.75 was obtained; the highest AUC was for prostate cancer AUC > 0.80. Linear and non-linear ML algorithms including socioeconomic, lifestyle, and clinical variables produced a moderate predictive performance of incident cancers in the Lifelines cohort.

**Abstract:**

Cancer incidence is rising, and accurate prediction of incident cancers could be relevant to understanding and reducing cancer incidence. The aim of this study was to develop machine learning (ML) models that could predict an incident diagnosis of cancer. Participants without any history of cancer within the Lifelines population-based cohort were followed for a median of 7 years. Data were available for 116,188 cancer-free participants and 4232 incident cancer cases. At baseline, socioeconomic, lifestyle, and clinical variables were assessed. The main outcome was an incident cancer during follow-up (excluding skin cancer), based on linkage with the national pathology registry. The performance of three ML algorithms was evaluated using supervised binary classification to identify incident cancers among participants. Elastic net regularization and Gini index were used for variables selection. An overall area under the receiver operator curve (AUC) <0.75 was obtained, the highest AUC value was for prostate cancer (random forest AUC = 0.82 (95% CI 0.77–0.87), logistic regression AUC = 0.81 (95% CI 0.76–0.86), and support vector machines AUC = 0.83 (95% CI 0.78–0.88), respectively); age was the most important predictor in these models. Linear and non-linear ML algorithms including socioeconomic, lifestyle, and clinical variables produced a moderate predictive performance of incident cancers in the Lifelines cohort.

## 1. Introduction

In European countries, the number of cancer incidence increased from 3.2 million in 2012 to 3.9 million in 2018. Breast, colorectal, lung, and prostate cancer were the most commonly diagnosed cancers [1]. These increased cancer rates are mainly related to the aging of the population, as over 50% of the new diagnosis of cancer are within people aged 65 years or older; elderly people aged 75+ account for more than one-third (36%) of cases, and the incidence in this group is expected to double by 2035 [2]. In addition, the implementation of cancer screening programs improved early detection and hence led to more cancers diagnosed [2,3] and the more prevalent lifestyle risk factors among the population [1,4]. Lifestyle risk factors that have an established association with the development of specific primary cancers are smoking [5], alcohol consumption [6], unhealthy diet [7,8], high body mass index (BMI) [9], low level of physical activity [10], and high level of sedentary behavior [11].

Thus far, predictions on the expected cancer incidence in a specified population are mainly extrapolated based on previously observed rates of cancer and age distribution within the population [12,13]. On the other hand, there are studies combining results from observed cancer rates in relation to lifestyle risk factors, estimating the effect of lifestyle modification on the expected incidence of cancer [14,15,16]. However, those estimated numbers are susceptible to bias, mainly because the combined impact of those lifestyle risk factors is dependent on population data available [12].

Recently it was shown that machine learning (ML) algorithms can be useful in the field of cancer prediction, since these algorithms are designed to identify complex or non-linear relationships between variables. These algorithms can also incorporate a large number of variables, which may increase model complexity and predictive performance compared to other statistical approaches [17,18,19,20,21,22,23]. The goal of these efforts was clearly to improve the early detection and prediction of incident cancers, prognosis, and survival [21,22,23]. However, there are some doubts as to whether ML algorithms indeed will perform better than traditional methods [24]. Moreover, it is suggested that scenarios showing a better performance of ML algorithms compared to commonly used statistics might have a very high risk of bias in the validation procedures, in addition to other limitations (i.e., relatively small sample size, limited number of predictors, or limited information about handling missing data) [24]. Furthermore, when structured clinical data are used to predict an incident cancer, the predictive performance of statistical methods and ML algorithms tends to be moderate to low [25]. As a result of this controversy, it is of importance to clarify if ML algorithms can demonstrate better performance in predicting an incident cancer when the above-mentioned limitations are overcome and to better understand how these predictions are accomplished. Considering this, the aim of the present study was to develop a model based on lifestyle, socioeconomic, and routine clinical variables to predict incident cancers in a prospective population-based cohort by using linear and non-linear ML algorithms.

## 2. Materials and Methods

### 2.1. Study Design

The present study used data from 167,729 participants collected between 2006 and 2013 from Lifelines, which is a multi-disciplinary prospective population-based three-generation designed cohort with 10% of the population from the northern Netherlands [26]. Participants were asked to fill in several structured and validated self-administered questionnaires about lifestyle, health status, and demographics. For a complete overview of the collected data, please see https://catalogue.lifelines.nl/ (accessed on 22 March 2021) [27]. The study protocol was approved by the medical ethics review committee of the University Medical Center Groningen [28]. Cancer diagnoses were provided by the linkage of the Lifelines database to the PALGA Foundation (Pathologisch-Anatomisch Landelijk Geautomatiseerd Archief) database. The PALGA Foundation database contains the registry of histopathology and cytopathology in the Netherlands and has nationwide coverage since 1991.

### 2.2. Participants

In the here presented analysis, we included adult participants (aged ≥ 18) from the Lifelines baseline assessment. If a participant was known with any type of cancer according to PALGA at the baseline assessment, they were excluded. In addition, participants with skin cancer diagnosis were excluded from the analysis, since this type of cancer might have different environmental causes than the ones evaluated in our analyses (i.e., UV exposure or radiation) [16].

### 2.3. Patient Outcome

The primary outcome was defined as any cancer diagnosis as provided by PALGA (date and location of cancer) after the date participants filled in the Lifelines baseline questionnaire (linkage of the data between PALGA and Lifelines was provided in September 2019). The outcome was further stratified as follows: all cancer types, and the most common cancer subtypes in Europe and among Lifelines participants: breast cancer, gastrointestinal cancers (colorectal, stomach, and esophagus, since they are considered to share similar lifestyle risk factors [8]), and prostate cancer.

### 2.4. Predictive Variables

Lifestyle, clinical, and socioeconomic variables extracted from the Lifelines database were used to predict outcome, as summarized in Appendix A. Variables with more than 30% missing values were excluded from the analysis, resulting in a total of 71 variables. Socioeconomic status was indicated by the level of education, which is commonly used for this purpose because it is easy to obtain, is more likely to have a low percentage of missing data, can be self-reported, and correlates well with other indicators of social stratification [29]. Education was classified as follows: low (i.e., no education, primary education, lower or preparatory vocational education, or lower general secondary education), medium (i.e., intermediate vocational education or apprenticeship, higher general senior secondary education, or pre-university secondary education), and high (i.e., higher vocational education or university). BMI was calculated by dividing the participant’s weight by the square of her or his height (kg/m^2^). Diet components were assessed with the 110-item food-frequency questionnaire (FFQ) that measured food intake over the previous month. The answers on these items were converted to grams per day using the method advised by the food-based Lifelines Diet Score [30]. Alcohol intake was also assessed by the FFQ. Physical activity was measured by the Short Questionnaire to Assess Health-enhancing Physical Activity (SQUASH) [31]. This questionnaire includes questions on routine activities (e.g., commuting, leisure, household, work, and school) and has been validated in the Dutch population. Only moderate to vigorous activities (expressed as total hours per week) in commuting and leisure time were considered for this analysis, because these domains better represent health behavior than occupational physical activity [32]. Sedentary behavior was evaluated by the total number of hours the participant spent watching TV per day. Smoking was expressed as the number of packages that a person consumed per year, by calculating the total grams of tobacco currently smoked, using the following equivalence: 1 cigarette = 1 g, one cigarillo = 3 g, and one cigar = 5 g [33] and then expressed as the total packages per year.

### 2.5. Statistical Analysis

Since the final dataset was highly imbalanced (the ratio of participants with a cancer diagnosis and without a history of cancer was low, see Table 1), and this can severely bias the performance of ML algorithms [34], several steps on analysis were undertaken to address this. First, a sample-size equalization strategy was applied in two different ways: (a) new cancer cases were matched to controls by sex, age, and education level; (b) a random selection of the participants with no history of cancer (only female participants for breast cancer model; only male participants for prostate cancer model); this resulted in multiple balanced datasets (50% cancer cases/50% no history of cancer) for every proposed scenario (all types of cancer, breast, prostate, and gastrointestinal cancer). Second, to evaluate the effect of aging, datasets were stratified for separate models by groups of age (i.e., 18–39; 40–49; 50–59; 60–69; and older than 70). That was done because the models matched by age, sex, and education level did not include these variables in their analysis.

### 2.6. Model Development and Validation

The performance of three common ML algorithms was evaluated to predict an incident cancer case for every balanced dataset (random selection, case-control, and stratified age categories) by supervised binary classification: (i) logistic regression, (ii) random forest, and (iii) support vector machines. Those algorithms have the ability to predict, either by linear (logistic regression) or non-linear approaches (random forest [35], support vector machines) a binary outcome and are the most frequently applied when using structured clinical data in cancer prediction [25]; this might be explained because they are computationally less expensive in comparison to other methods [21] (i.e., deep learning) but still with a high predictive performance, and they also are flexible over the possible distributions of the data included. In addition, the predictive factors can be directly obtained from the algorithms [21,22,23,25] (for further details about the algorithms, see Appendix A). For all the datasets, the same model architecture and modeling procedure was followed (see Figure 1). The first step was to randomly split the data and to use 80% of the data for training and the remaining 20% for testing. Then, missing data in the training set were estimated through multiple imputation by chained equations (MICE) [36], setting five rounds of imputation and replacing the missing values with the fifth round for all models, using the “mice” package in R statistics. After imputation, all the variables in the training set were standardized by using uniform means and standard deviations (subtracted by the mean and divided by the standard deviation). Missing values in the testing set were estimated separately also using MICE; in addition, standardization of the variables in the testing set was calculated using the mean and standard deviation of the training set.

The predictive performance of the algorithms in the training set was evaluated using a 10-fold cross-validation technique. In every fold, data from 9/10 folds were used for the training of the prediction model, and the remaining one was used for testing; this was repeated until all folds in the training set had been used. During classification, the following hyperparameters were tuned: (a) for random forest, using the ranger function in the caret package: (i) number of randomly selected variables for each split (mtry), (ii) number of trees (200, 300, 500), and (iii) minimum node size (0, 0.5, and 1); (b) for support vector machines a linear kernel was used, and the adjusted hyperparameter was the cost of constraints violation “C”, searched from 0 to 2 in steps of 0.5.

Initially, all the variables were included in the classification process. In the next step, only variables relevant for incident cancer prediction were used (see Figure 1). To identify these important variables, two different techniques were used. First, the mean decrease Gini index criteria (MDG) scales the importance of the variables within the algorithm in a range from 0 to 100. Within that range, as a standard procedure, the most important 20 variables in the Gini scale from the initial random forest classifier were considered for the final models. Second, the elastic net regularization was applied, and variables with coefficients equal to zero were not included in the further models [37] (see Figure 1). A heat map was generated to show the most important variables across the models. As a final step, the prediction ability of every model was evaluated in the testing set. The area under the receiver operator characteristic curve (AUC) was the performance metric, a higher AUC value indicates a better performance of the model (where an AUC value of 1 equals optimal performance and an AUC value of 0.5 is considered low predictive performance). To improve transparency, the model development followed the “transparent reporting of a multivariable prediction model for individual prognosis or diagnosis” (TRIPOD) statement; these guidelines were established to improve the reliability when developing and reporting prediction models and might help to address limitations from previous studies, to produce fair comparisons among models and more reliable metrics [38,39]. All analyses were conducted in R statistics, version R-3.5.2, with the ‘Caret’ package. All results are reported as mean AUC (±95% CI), unless stated otherwise.

## 3. Results

A total of 120,420 participants were included in the study, out of which 4232 had a new cancer diagnosis (see Table 1). The majority of participants were females (58.34%, *n* = 70,260); those participants without a history of cancer had a mean age of 43.62 years (SD 12.68). The mean age of the of the new detected cancer cases was 52.53 (SD 13.12); there was a higher prevalence of new cancer incidents among females (61.0% *n* = 2 581). From the participants who were diagnosed with a new cancer, 74.3% (*n* = 3 147) reported a low or medium socioeconomic status. The most incident cancer cases in the follow up were breast cancer (*n* = 977), followed by gastrointestinal (*n* = 609) and prostate (*n* = 508) (see Table 2).

### 3.1. General Model

In the models analyzing all types of cancer, including those stratified by age subgroups, the AUC was below 0.70 (see Appendix A). The best performance of the three algorithms was achieved in the model with random forest variable selection and random controls (random forest AUC = 0.65 (95% CI 0.62–0.67), logistic regression AUC = 0.66 (95% CI 0.63–0.68) and support vector machines AUC = 0.65 (95% CI 0.63–0.67), respectively). However, this was not substantially higher when compared to the models that included all the variables (random forest AUC = 0.64 (95% CI 0.62–0.67), logistic regression AUC = 0.64 (95% CI 0.62–0.66) and support vector machines AUC = 0.63 (95% CI 0.61–0.66), respectively). The variables that added the most to these models were age (MDG = 100), savory and ready product intake (MDG = 26.26), summed polysaccharide intake (MDG = 17.13), granulocyte count (MDG = 16.66) and number of smoking pack years (MDG = 15.69) see Figure 2.

### 3.2. Breast Cancer Model

The breast cancer models had an AUC below 0.70 (Appendix A). The highest predictive performance was achieved by the model for age group 70+ years with random forest variable selection (random forest AUC = 0.68 (95% CI 0.44–0.92), logistic regression AUC = 0.56 (95% CI 0.30–0.82), and support vector machines AUC = 0.56 (95% CI 0.30–0.82), respectively). Variables with high MDG in these models were number of television hours per day (MDG = 100), eosinophils count (MDG = 94.97), cholesterol LDL (MDG = 82.69), lymphocytes count (MDG = 77.00) and refined grain intake (MDG = 74.47) (see Figure 2). In the models with random controls, age groups 40–49, 50–59, 60–69, and 70+, it was not possible to develop models with elastic net variable selection, because no relevant variables were identified (Appendix A).

### 3.3. Gastrointestinal Cancer Model

The performance of the gastrointestinal cancer models was below AUC = 0.75 (Appendix A). The best performing algorithm was based on random forest variable selection and random controls (random forest AUC = 0.71 (95% CI 0.65–0.77), logistic regression AUC = 0.75 (95% CI 0.69–0.80), and support vector machines AUC = 0.72 (95% CI 0.66–0.77), respectively); nevertheless, it was not considerably higher compared to the model that included all variables (random forest AUC = 0.73 (95% CI 0.68–0.79), logistic regression AUC = 0.72 (95% CI 0.66–0.77), and support vector machines AUC = 0.73 (95% CI 0.67–0.79), respectively). In these models, the most contributing variables were age (MDG = 100), number of smoking pack years (MDG = 24), savory and ready product intake (MDG = 22.42), glycosylated hemoglobin level (MDG = 19.73), BMI (MDG = 13.36), and triglyceride level (MDG = 11.38).

### 3.4. Prostate Cancer Model

In the prostate cancer models, all the case controls scenarios had an AUC below 0.70 (Appendix A), and the highest AUC for all the algorithms was achieved when using random controls, either by the inclusion of either optimal features by random forest variable selection (random forest AUC = 0.82 (95% CI 0.77–0.87), logistic regression AUC = 0.81 (95% CI 0.76–0.86), and support vector machines AUC = 0.83 (95% CI 0.78–0.88), respectively) or by including all variables (random forest AUC = 0.82 (95% CI 0.77–0.87), logistic regression AUC = 0.76 (95% CI 0.70–0.82), and support vector machines AUC = 0.80 (95% CI 0.74–0.85), respectively). Variables that contributed most to these models were age (MDG = 100), savory and ready product intake (MDG = 14.62), creatinine level (MDG = 8.07), glycosylated hemoglobin level (MDG = 8.07), and sugar beverage intake (MDG = 6.23) (see Figure 2).

## 4. Discussion

The aim of the present study was to develop models to predict an incident cancer case in a prospective population-based cohort, by using linear and non-linear algorithms to identify the most contributing variables in these predictions. This is important to gain insight into factors contributing to cancer development, which might become future targets for cancer prevention. The models for the development of cancer in general and breast cancer showed low predictive performance. On the other hand, gastrointestinal cancer showed moderate predictive performance. The highest predictions obtained in this study were achieved when random controls were included, and age was predominantly the most contributing variable within those models. The predictive performance of the models increased to some extent when performing variable selection. The best predictive performance was obtained in the prostate cancer model when incident cases were matched to random controls (AUC > 0.80). Age, savory-ready product intake, creatinine levels, glycosylated hemoglobin levels, and sugar beverage intake were the most contributing variables in these models.

Age was a strong predictor in the unmatched models, which is in line with the expectation, as cancer is mainly a disease of aging [40], where older age is responsible for half of the diagnoses irrespective of the type of cancer [2]. As such, age obscured the effect of the other explanatory variables included. However, it is known that etiology differs between cancer types; because of this, it is implicit that prediction models generally perform better when designed for a specific cancer type [21]. To our knowledge, no predictive models for general incident cancer prediction using machine learning algorithms have been described in the literature. Therefore, the performance of our general incident cancer prediction model could not be compared to other models. However, it was possible to derive that lifestyle and clinical variables produced a low predictive performance in overall cancer incidence prediction. In addition, variable selection using (ML) algorithms only slightly improved the predictive performance in this particular scenario [41].

Breast cancer prediction, on the other hand, has been widely assessed [42,43,44], and our models in this study showed comparable performance to those reported in a recent review [44] where the AUC is usually <0.70 when genetic data are not included. Conversely, a recent study showed a striking predictive performance of cancer when using ML algorithms compared to the breast cancer risk assessment tool (BCRAT) and the breast and ovarian analysis of disease incidence and carrier estimation algorithm (BOADICEA) models, using the same variables (AUC > 0.90), although this is an analysis that includes specific genetic variations for breast cancer prediction. [20]. Although the predictive performance is high, the procedure to compare the models might be biased or not so fair; since it did not use a separate testing set to blind the outcome, ML algorithms used supervised classification and rebalanced datasets during training. On the contrary, that study received serious criticism about the risk of bias in the validation procedure and the fair comparisons between the models [45]. In contrast, the here-presented approach used the same variables and the same structure to train the algorithms, used a separate testing set to blind the outcome achieve internal validation. The availability of studies focused on prediction of breast cancer by lifestyle, socioeconomic, and clinical variables using ML approaches is scarce [44]. However, our results are in line with those reported for low biased breast cancer prediction models using ML (AUC < 0.70) [42,43,44]. The observed limited performance is likely due to the relevance of the variables included, as well as the fact that genetic data were not included. In our models, the predictive performance increased when Gini index variable selection was used.

The optimal gastrointestinal cancer models showed moderate predictive performance. Although the association between diet and specific cancer risk has been clearly established in previous studies [8], the here-presented analysis focused on the prediction and not the risk assessment, which makes it difficult to compare these studies. The food groups included here were evaluated separately and not included as diet indexes/diet scores [46], which might have an impact on the prediction, because the combined effect of the components could not be assessed in our approach.

The prostate cancer model showed the highest predictive performance for the optimal models (AUC > 0.80), which is similar to the results obtained in a study using random forest in which an AUC > 0.80 was achieved [47]. In this study, it was also reported that age was the most contributing variable. One of the main differences with this previous study was their inclusion of specific biomarkers in the prediction model, where our study only included lifestyle, socioeconomic, and clinical variables. However, both studies achieved comparable performance. To further strengthen the results of this study, external validation of this prostate cancer model should be performed.

Recently published studies have been optimistic about the performance of cancer prediction models, claiming that non-linear ML algorithms may outperform linear-based statistical methods [23,48]. Nevertheless, it is of concern that such models could lead to model overfitting or face a lack of interpretability by prioritizing prediction over explanation [48]. The results from the present study showed that the predictive performance of non-linear ML algorithms and linear-based methods are similar in this specific scenario, if previous limitations (relatively small sample size, limited number of predictors, improper internal validation, hyperparameter tuning, and no feature selection) are addressed by adhering to reporting guidelines. This overall moderate or low performance might be explained by the use of tabular structured data, or the use of lifestyle, socioeconomic, blood, and urine test variables, but also because genetic data or specific biomarkers were not included in the present analysis [25] and as such, the number of variables was limited. In addition, the overall findings are in line with the results from a systematic review in which low-bias clinical prediction models showed no differences in performance when comparing ML algorithms to logistic regression [24].

Some strengths in the present study need to be mentioned: First, our study adhered to the recently proposed guidelines to develop and improve the authenticity when reporting prediction models (TRIPOD statement) [38]. Second, previous limitations stated in the literature such as sample size, reduced number of predictors, proper internal validation, hyperparameters tuning, feature selection, and an additional case control analysis were addressed to reduce bias in the prediction models. Third, cancer cases were retrieved from a nationwide pathology registry. Fourth, this study included a large number of predictors from several health domains. Fifth, the dataset had a low number of missing values, and those occurring were imputed according to reporting guidelines. Finally, a comprehensive assessment of variables and check of their reliability was performed, in order to reduce bias or measurement errors. In a study with these strong points, the predictive performance of non-linear ML algorithms and linear-based methods are similar in this specific scenario.

In addition, several limitations need to be mentioned: first, the models did not incorporate time to event of diagnosis, and they only considered one time measurement at baseline to make a prediction; this might impact the results, since multiple measurements might better show exposure to the predictors and therefore increase the predictability. Second, genetic data were not included, since genetic data were not available for all the participants included in this analysis; the inclusion of genetic data might also increase the predictive performance of ML algorithms. Third, the analysis is only based on baseline assessment of the predictors; no follow-up data were included; in the future, the inclusion of follow-up data might improve the performance of the models. Fourth, the included variables are seen as raw data; only smoking and physical activity were included as pack years smoked and total hours of moderate and vigorous physical activity; thus, the combined effect of some variables (i.e., diet scores) might not be reflected in the outcome. Fifth, since the present study only evaluated in separate models the most common cancer diagnosis in the Lifelines cohort, future research might evaluate independently other subtypes of cancer for which this study did not have a large enough sample size.

What could be the implications of our findings? As to clinical applicability, the moderate or high predictive performance in the prostate and GI cancer models is promising, but this must be externally validated to derive possible clinical relevance. Of note, it is important to emphasize that causality is not assumed in our models. This is because the proposed ML algorithms did not include specific biomarkers and only focused on the predictive effect of socioeconomic factors, lifestyle behaviors, and routine clinical variables on a cancer outcome. For the exploration of possible causal factors, further analyses in the current and other population-based cohorts are warranted. That could focus, for instance, on the savory snacks and ready-to-eat products as well as glycosylated hemoglobin, which we found to be among the most important variables in both prostate and GI cancer models.

## 5. Conclusions

In conclusion, the performance of linear and non-linear ML algorithms for predicting all incident cancers, breast cancers, and gastrointestinal cancers using socioeconomic, lifestyle, and clinical variables from the Lifelines cohort is moderate to low, where age is the most important predictor. The model developed to predict incident prostate cancer, which included all the variables, achieved high predictive performance. Explanations can be that the data included to make these predictions are relatively homogeneously distributed. Further studies might be able to include other types of data such as genetic, which might help to improve the predictive performance of incident cancer cases.

## Figures and Tables

**Figure 1 cancers-13-02133-f001:**
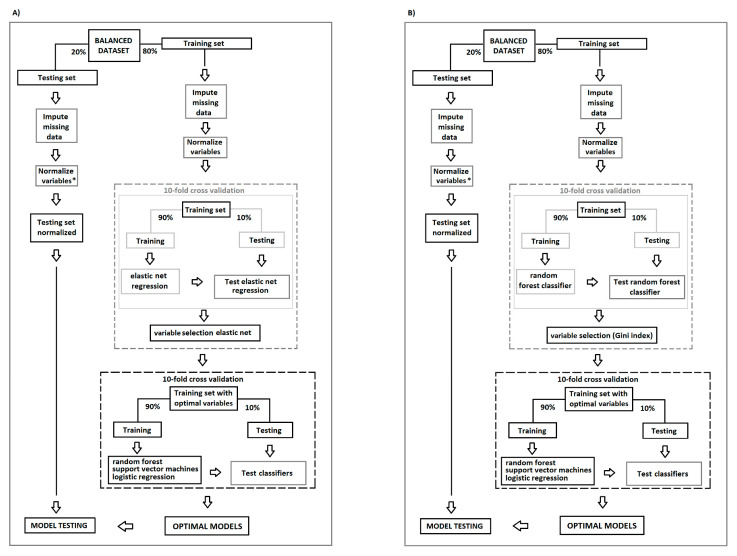
Data analysis process, variable selection strategies: (**A**) by elastic net regression and (**B**) Gini index, and cross-validation performance to obtain the optimal models for cancer prediction. * Variables in the testing set were standardized using the mean and standard deviations from the training set.

**Figure 2 cancers-13-02133-f002:**
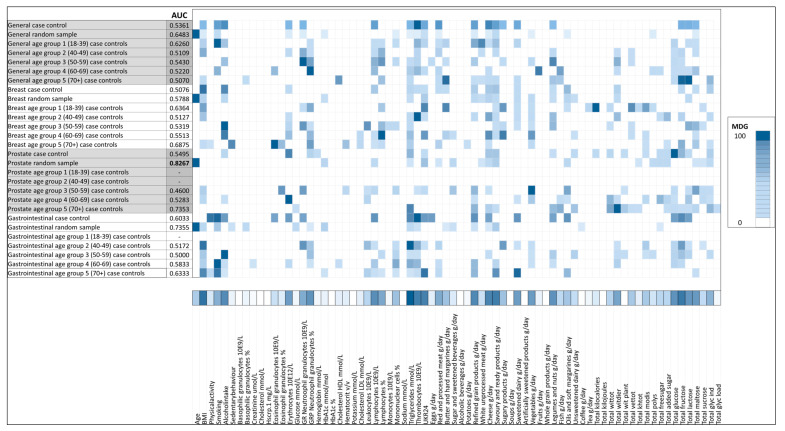
Area under the receiver operator curve (AUC), for random forest models and the most important variables for classification based on the Gini index (MDG).

**Table 1 cancers-13-02133-t001:** Baseline characteristics of the 120,420 participants stratified by those who had a new cancer diagnosis in the follow-up versus those without any history of cancer (for age and sex-matched controls and the others without any history of cancer).

Variation	Cancer in Follow-Up	Controls	Without Any History of Cancer
	(*n* = 4232)	(*n* = 4232)	(*n* = 116,188)
Baseline age (SD)	52.53 (13.12)	52.53 (13.12)	43.62 (12.68)
Sex			
Females (%)	2581 (61.0%)	2581 (61.0%)	67,679 (58.2%)
Education level			
Low (%)	1715 (40.5%)	1658 (39.2%)	34,678 (29.8%)
Medium (%)	1432 (33.8%)	1418 (33.5%)	46,238 (39.8%)
High (%)	1085 (25.6%)	1156 (27.3%)	35,272 (30.4%)
Baseline body mass index (SD)	26.62 (4.22)	26.31 (4.30)	25.97 (4.30)
Baseline alcohol intake grams/day (SD)	8.12 (9.69)	7.15 (8.67)	7.20 (8.93)
Smoking packages/year (SD)	10.20 (13.89)	7.92 (11.83)	5.90 (9.57)
Baseline physical activity h/week (SD)	4.51 (5.49)	4.46 (5.14)	4.14 (4.80)
Baseline sedentary behavior TV h/day (SD)	2.70 (1.53)	2.66 (1.53)	2.46 (1.48)

SD: standard deviation; h: hours; TV: television.

**Table 2 cancers-13-02133-t002:** Characteristics of the participants for the case control analysis stratified by cancer diagnosis.

	Breast Cancer in Follow-Up	Breast Cancer Random Controls	Breast Cancer Controls	Prostate Cancer in Follow-Up	Prostate Cancer Random Controls	Prostate Cancer Controls	GI Cancer in Follow-Up	GI Cancer Random Controls	GI Cancer Controls
	(*n* = 977)	(*n* = 977)	(*n* = 977)	(*n* = 508)	(*n* = 508)	(*n* = 508)	(*n* = 609)	(*n* = 609)	(*n* = 609)
Baseline age (SD)	50.76 (10.60)	43.74 (12.83)	50.76 (10.60)	62.75 (7.48)	45.08 (12.70)	62.75 (7.48)	57.22 (10.73)	44.68 (12.01)	57.22 (10.73)
Sex									
Females (%)	977 (100%)	977 (100%)	977 (100%)	-	-	-	249 (40.9%)	361 (59.2%)	249 (40.9%)
Education level									
Low (%)	390 (39.9%)	305 (31.2%)	364 (37.3%)	222 (43.7%)	166 (32.7%)	228 (44.9%)	274 (45.0%)	177 (29.1%)	274 (45.0%)
Medium (%)	345 (35.3%)	392 (40.1%)	365 (37.4%)	134 (26.4%)	198 (39.0%)	131 (25.8%)	189 (31.0%)	258 (42.4%)	171 (28.1%)
High (%)	242 (24.8%)	280 (28.7%)	248 (25.4%)	152 (29.9%)	144 (28.3%)	149 (29.3%)	146 (24.0%)	174 (28.5%)	164 (26.9%)

Body mass index (SD)	26.47 (4.53)	25.74 (4.81)	26.10 (4.57)	26.72 (3.19)	26.18 (3.18)	27.04 (3.53)	27.47 (4.15)	25.91 (4.06)	26.51 (3.82)
Alcohol intake grams/day (SD)	5.97 (7.45)	4.75 (5.88)	5.10 (6.41)	10.93 (9.74)	10.39 (11.35)	10.47 (10.16)	9.58 (10.95)	7.17 (8.89)	8.19 (8.89)
Smoking packages/year (SD)	6.46 (9.66)	5.36 (9.03)	6.53 (9.46)	11.66 (13.87)	7.35 (10.44)	12.52 (14.97)	13.14 (16.60)	5.84 (9.15)	9.19 (12.97)
Physical activity h/week (SD)	4.26 (4.90)	4.30 (5.01)	4.35 (5.05)	5.45 (6.46)	4.40 (5.20)	5.32 (6.27)	4.73 (5.92)	4.65 (5.80)	5.06 (6.10)
Sedentary behavior TV h/day(SD)	2.66 (1.41)	2.42 (1.32)	2.63 (1.33)	2.80 (1.70)	2.42 (1.23)	2.78 (1.37)	2.84 (1.48)	2.44 (2.09)	2.67 (1.31)

GI: gastrointestinal; SD: standard deviation; TV: television; h: hours.

## Data Availability

The data presented in this study are available on request from the corresponding author. The data are not publicly available due data protection from the Lifelines Biobank, see https://www.lifelines.nl/ (accessed on 22 March 2021).

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
