# Peer review of "Prediction of Incident Cancers in the Lifelines Population-Based Cohort"

_cancers, 2021, doi:10.3390/cancers13092133_

Round 1

Reviewer 1 Report

The authors have addressed my concerns.  I think this paper is a nice early addition to the literature on the many uses of machine learning.  I look forward to similar results from larger datasets looking at more cancers.

This manuscript is a resubmission of an earlier submission. The following is a list of the peer review reports and author responses from that submission.

Round 1

Reviewer 1 Report

The machine learning methods appear to be carried out with care, but a great deal of information is missing:  How much imputation was needed and for what variables?

Instead of titling this Prediction of incident cancers”….I suggest they focus on prostate cancer and mention that their algorithms did not perform well for breast cancer.  Combining GI cancers is not a good idea as it is clear that the etiologies for each of those is quite different and in many cases unknown.  It would have been interesting to include the risk factor information for each of those and for lung in the supplementary data.  These are all solid tumors; what about the blood cancers?

Due to these large errors, this reviewer did not comment on small modifications that might have made the paper more readable.

Author Response

Point-by-point response: Reviewer 1

Comment 1

The machine learning methods appear to be carried out with care, but a great deal of information is missing:  How much imputation was needed and for what variables?

Reply: To better show how much imputation was needed, we included in the supplementary file (Table S1) additional information about the percentage of missing values per variable, we mentioned in the methods section from the original version of our manuscript that only variables with less than 30% of missing values were included. In this way, we precise the percentage of missing values per variable. It is also important to clarify that during the imputation process (using MICE package in R) we set five rounds of imputation per variable and used the 5th imputation round to replace the missing values. Because of this we added the following text to the methods section 2.6 as highlighted:

‘Then, missing data in the training set were estimated through multiple imputation by chained equations (MICE) [1], setting five rounds of imputation and replacing the missing values with the fifth round for all models, using the “mice” package in R statistics’

Comment 2

Instead of titling this Prediction of incident cancers”….I suggest they focus on prostate cancer and mention that their algorithms did not perform well for breast cancer

Reply: Thank you for this recommendation, however we thought about our title as a match with our main hypothesis and research question to evaluate the predictive performance for all cancers and the most common cancers in the Lifelines cohort (breast, prostate and gastrointestinal cancers) ) to make our analysis more specific. The main rational for this is that it has been suggested that general factors like lifestyle factors are associated with an increased risk for cancer in general. So, the main aim to this study was to evaluate which general factors, besides ageing, are related to the incidence of cancer, we prefer to stick to this more general title.

Comment 3

Combining GI cancers is not a good idea as it is clear that the etiologies for each of those is quite different and in many cases unknown. It would have been interesting to include the risk factor information for each of those and for lung in the supplementary data.

Reply: We agree that there are many subtypes of cancer, and the group GI cancers consist also out of many subtypes. In the here presented analyses, we assumed that the occurrences of these cancer share some risk factors that are mainly related to lifestyle factors. As recent evidence suggest that an unhealthy diet commonly represents a higher risk for esophageal, stomach and colorectal cancers [2]. To make this more clear we added the following to the methods section.

‘The outcome was further stratified as follows: all cancer types, and the most common cancer subtypes in Europe and among Lifelines participants: breast cancer, gastrointestinal cancers (colorectal, stomach and esophagus, since they are considered to share similar lifestyle risk factors [2]) and prostate cancer.’

Comment 4

These are all solid tumors; what about the blood cancers?

Reply: We are aware that we only performed separate models for the most common incident cancers in the Lifelines cohort (breast, prostate and gastrointestinal ), they accounted for 50% of the total diagnoses, which is in line with the most common cancers diagnosed in the Netherlands. Due to the limited amount of cases for the rest of the diagnoses, we decided to focus in those three. But we believe that this might be an opportunity for future research to be made in specific types of cancer that we did not describe in our own study, when enough sample size is achieved. We added the following text to the limitations part of our discussion to better address this limitation:

‘Fifth, since the present study only evaluated in separate models the most common cancer diagnosis in the Lifelines cohort, future research might evaluate independently other subtypes of cancer for which this study did not have enough sample size. ’

References

[1]        White IR, Royston P, Wood AM. Multiple imputation using chained equations: Issues and guidance for practice. Stat Med 2011;30:377–99. https://doi.org/10.1002/sim.4067.

[2]        Grosso G, Bella F, Godos J, Sciacca S, Del Rio D, Ray S, et al. Possible role of diet in cancer: systematic review and multiple meta-analyses of dietary patterns, lifestyle factors, and cancer risk. Nutr Rev 2017;107:1233–9. https://doi.org/10.1093/nutrit/nux012.

Reviewer 2 Report

This is a well written paper that offers some insight about the comparison among three predictive models with varying statistical assumptions.  The results are consistent with the current literature.  There are several critical issues that need to be addressed.  First, because the age effect is so dominant and predictive of incident cancers, it may be more fruitful that authors could document more evidence from the literature review of age effects on the incidence of cancer.  Perhaps, one feasible approach could be undertaken to tease out the net effect of age on cancer incidents, using a propensity score matching and analytical approach.  Second, it is unclear about the theoretical assumptions in applying three different statistical estimation methods in the analysis. Third,  the Lifelines population is a unique cohort that have offered the opportunity to perform more precise statistical analysis.  For example,  the authors could take the advantage of the cohort design, using time-varying predictors in the investigation.  I recognize the authors noted the limitations for the lack of attention to time-varying predictors in modeling.  Fourth,  age stratification is helpful in examining the variations in incidents of varying cancer.  However, it is unclear that this approach will reveal any new information about age curves or distributions of cancer incidents, irrespective of the type of cancer.

Author Response

Point-by-point response: Reviewer 2

 Comment 1

This is a well written paper that offers some insight about the comparison among three predictive models with varying statistical assumptions. The results are consistent with the current literature.

Reply: Thank you for these kind words.

Comment 2

There are several critical issues that need to be addressed. First, because the age effect is so dominant and predictive of incident cancers, it may be more fruitful that authors could document more evidence from the literature review of age effects on the incidence of cancer.

Reply: We agree with the reviewer to add more evidence from literature, and to better document the effects of ageing on the incident cancer cases, we changed the following text and added the references in the first paragraph of the introduction. Now it reads in the introduction:

These increased cancer rates are mainly related to the ageing of the population, as over 50% of the new diagnosis of cancer are within people aged 65 years or older, and elderly people aged 75+ account for more than a third (36%) of cases, and the incidence in this group is expected to double by 2035 [1]

we have also added the following statement in the discussion section as highlighted:

‘Age was a strong predictor in the unmatched models, which is in line with the expectation, as cancer is mainly a disease of ageing [2], where older age is responsible for a half of the diagnoses irrespective of the type of cancer [1].’

Comment 3

Perhaps, one feasible approach could be undertaken to tease out the net effect of age on cancer incidents, using a propensity score matching and analytical approach.

Reply: We thank the reviewer for this suggestion, and we agree that there are several ways to address the problem associated with imbalanced datasets. Since the primary focus of this study was developing a system for prediction of incident cancers using multiple factors ( instead of the evaluation of one specific factor), we decided that the propensity score method was out of the scope in this study. Instead we applied a more general approach with the application of machine learning algorithms to explore which factors, other than age, are potential predictor for incident cancer in this database.

Comment 4

Second, it is unclear about the theoretical assumptions in applying three different statistical estimation methods in the analysis.

Reply: We agree that the theoretical assumptions behind the selection of machine learning algorithms was not clear. As such, we have added brief description on the theoretical assumptions in the methods section of the revised manuscript and included further details on a supplementary file, now it reads as follows:

‘Those algorithms have the ability to predict, either by linear (logistic regression) or non-linear approaches (random forest [3], support vector machines) a binary outcome and are the most frequently applied when using structured clinical data in cancer prediction [4], this might be explained because they are computationally less expensive in comparison to other methods [5](i.e. deep learning) but still with a high predictive performance, also are flexible over the possible distributions of the data included. In addition, the predictive factors can be directly obtained from the algorithms [4–7] (for further details about the algorithms see supplementary file S3)’

Comment 5

Third, the Lifelines population is a unique cohort that have offered the opportunity to perform more precise statistical analysis. For example, the authors could take the advantage of the cohort design, using time-varying predictors in the investigation. I recognize the authors noted the limitations for the lack of attention to time-varying predictors in modelling

Reply: We agree with the reviewer that an analytic approach might have been an option to analyse this database. However, we in this study we decided to go for a holistic approach, as these approaches are very promising and are capable to use cross-sectional data to build a prediction model, as was our original design for our study. However, this approach also has limitations. One is that time-to event component of data cannot be analysed within this strategy, as has been mentioned in the discussion.

 Comment 6

Fourth, age stratification is helpful in examining the variations in incidents of varying cancer.  However, it is unclear that this approach will reveal any new information about age curves or distributions of cancer incidents, irrespective of the type of cancer.

Reply: the age stratification was applied in our case-control datasets, since we expected that the effect of our lifestyle, clinical and socioeconomic variables would be different for the age groups in the case-control analysis. However, we observed that the AUC´s for the older groups had very similar predictive performance compared to the younger groups. The possible explanation for this was the homogeneity in Lifelines regarding lifestyle, clinical and socioeconomic variables.

References

[1]        Pilleron S, Sarfati D, Janssen-Heijnen M, Vignat J, Ferlay J, Bray F, et al. Global cancer incidence in older adults, 2012 and 2035: A population-based study. Int J Cancer 2019;144:49–58. https://doi.org/10.1002/ijc.31664.

[2]        De Magalhães JP. How ageing processes influence cancer. Nat Rev Cancer 2013;13:357–65. https://doi.org/10.1038/nrc3497.

[3]        Breiman L. Random Forests. Mach Learn 2001:5–32. https://doi.org/10.14923/transinfj.2015IUP0008.

[4]        Richter AN, Khoshgoftaar TM. A review of statistical and machine learning methods for modeling cancer risk using structured clinical data. Artif Intell Med 2018;90:1–14. https://doi.org/10.1016/j.artmed.2018.06.002.

[5]        Kourou K, Exarchos TP, Exarchos KP, Karamouzis M V., Fotiadis DI. Machine learning applications in cancer prognosis and prediction. Comput Struct Biotechnol J 2015;13:8–17. https://doi.org/10.1016/j.csbj.2014.11.005.

[6]        Cruz JA, Wishart DS. Applications of Machine Learning in Cancer Prediction and Prognosis. Cancer Inform 2006;2:59–78. https://doi.org/10.1177/117693510600200030.

[7]        Huang S, Yang J, Fong S, Zhao Q. Artificial intelligence in cancer diagnosis and prognosis: Opportunities and challenges. Cancer Lett 2020;471:61–71. https://doi.org/10.1016/j.canlet.2019.12.007.

Round 2

Reviewer 1 Report

I appreciate the effort that the authors have gone to to respond to comments.  However, this paper needs some clinician input as it seems quite "vague" in terms of results.  I realize that the authors are using ML, but it is rather difficult to see the utility of ML in this context.  What have we learned from this exercise?  This is not clear to the reader.

Reviewer 2 Report

The responses have clarified some of the concerns.  Given the availability of cross-sectional data, the analytical approach undertaken by authors is reasonable.  No further comments are suggested.  I recommend that the paper be further considered by the review panel and the editorial staff.